

# Impacts of land use change on native plant-butterfly interaction networks from central Mexico

Deysi Muñoz-Galicia[1], Carlos Lara[2], Citlalli Castillo-Guevara[2], Mariana Cuautle[2] and Claudia Rodríguez-Flores[3]

[1] Maestría en Biotecnología y Manejo de Recursos Naturales, Universidad Autónoma de Tlaxcala, San Felipe Ixtacuixtla, Tlaxcala, Mexico
[2] Centro de Investigación en Ciencias Biológicas, Universidad Autónoma de Tlaxcala, San Felipe Ixtacuixtla, Tlaxcala, Mexico
[3] Departamento de Conservación de la Biodiversidad, El Colegio de la Frontera Sur, Villahermosa, Tabasco, Mexico

Corresponding author
Carlos Lara,
carlos.lara.rodriguez@gmail.com

## ABSTRACT

**Background**. Land use change is a key catalyst of global biodiversity loss and ecosystem degradation. Deforestation and conversion of natural habitats to agricultural or urban areas can profoundly disrupt plant-flower visitor interactions by altering their abundances and distribution. Yet, specific studies analyzing the effects of land use change on the structure of networks of the interactions between particular groups of flower visitors and their plants are still scarce. Here, we aimed to analyze how converting native habitats affects the species composition of butterfly communities and their plants, and whether this, in turn, leads to changes in the structure of interaction networks in the modified habitats.

**Methods**. We performed bi-monthly censuses for a year to record plant-butterfly interactions and assess species diversity across three habitat types, reflecting a land-use change gradient. From original native juniper forest to urban and agricultural zones in central Mexico, one site per land use type was surveyed. Interactions were summarized in matrices on which we calculated network descriptors: connectance, nestedness and modularity.

**Results**. We found highest butterfly diversity in native forest, with the most unique species (*i.e.*, species not shared with the other two sites). Agricultural and urban sites had similar diversity, yet the urban site featured more unique species. The plant species richness was highest in the urban site, and the native forest site had the lowest plant species richness, with most of the plants being unique to this site. Butterfly and plant compositions contrasted most between native forest and modified sites. Network analysis showed differences between sites in the mean number of links and interactions. The urban network surpassed agriculture and native forest networks in links, while the native forest network had more interactions than the agriculture and urban networks. Native plants had more interactions than alien species. All networks exhibited low connectance and significant nestedness and modularity, with the urban network featuring the most modules (*i.e.*, 10 modules).

**Conclusions**. Converting native habitats to urban or agricultural areas reshapes species composition, diversity and interaction network structure for butterfly communities and plants. The urban network showed more links and modules, suggesting intricate urban ecosystems due to diverse species, enhanced resources, and ecological niches

encouraging interactions and coexistence. These findings emphasize the impacts of land use change on plant-butterfly interactions and the structure of their interaction networks.

## INTRODUCTION

Changes in land use and vegetation cover resulting from the expansion of anthropogenic activities have negative impacts on biodiversity and the provision of ecosystem services, and they contribute significantly to climate change processes at regional scales (*Davison, Rahbek & Morueta-Holme, 2021*). Over the past decade, numerous studies have warned of the impacts of land use change on pollinators around the world, particularly in terms of population decline and species extinction (*Potts et al., 2010*; *Burkle, Marlin & Knight, 2013*; *Senapathi et al., 2015*; *Dicks et al., 2021*). As animal pollination is a key ecological process for the reproductive success of most flowering plants, including those of agricultural interest, a pollinator deficit could have serious consequences for humanity (*Aizen & Harder, 2009*; *Melathopoulos, Cutler & Tyedmers, 2015*; *Marshman, Blay-Palmer & Landman, 2019*; *Jordan et al., 2021*).

Overall, changes in land use result in a reduction in natural habitats and an increase in the availability of anthropized habitats at the landscape scale (*Tilman et al., 2017*). This may lead to a decrease in the abundance and composition of pollinator communities by decreasing native floral resources and, possibly, increasing exotic flora (*Winfree, Bartomeus & Cariveau, 2011*). In this context, the impact of land use change on pollination networks (and systems with the same formal structure, such as host-parasitoid networks) can lead to the loss of interactions at the local scale, particularly affecting the most specialized interactions (*Tylianakis, Tscharntke & Lewis, 2007*; *Aizen, Sabatino & Tylianakis, 2012*; *Staniczenko et al., 2017*). As a consequence, this process may promote the formation of subgroups (modules) comprising species that are more tightly connected to each other than to species in other modules. Nevertheless, it is not entirely clear how this phenomenon contributes to an increase in modularity. Furthermore, such changes could potentially reduce the overall stability of these interaction networks (*Spiesman & Inouye, 2013*). In this regard, considering the growing demand for pollination services, more studies are being conducted to assess how the conversion of natural habitats to agricultural, livestock, and urban uses has affected the structure of plant-flower visitor interaction networks (*e.g., Yoshihara et al., 2008*; *Hagen & Kraemer, 2010*; *Vanberger et al., 2014*; *Moreira, Boscolo & Viana, 2015*; *Baldock et al., 2015*; *Baldock et al., 2019*; *Díaz-Infante, Lara & Arizmendi, 2020*).

Butterflies' lives are strongly intertwined with plants. As larvae, they need plant tissues to feed and develop (*i.e.,* they act as herbivores); as adults, they depend on floral nectar to survive (*i.e.,* they transfer pollen between plants to ensure their reproductive success);

and to reproduce, they must find suitable host plants to lay their eggs (*Snodgrass, 1961*). Thus, within this framework of mutual dependencies between plants and butterflies, land-use changes are expected to affect whole interaction networks (*Pearse & Altermatt, 2013*). Previous studies have shown that butterfly species with high habitat specialization (and low dispersal ability) are more vulnerable to extinction from land use change than butterfly species with low specialization (and high dispersal ability) (*Koh, Sodhi & Brook, 2004*; *Bergerot et al., 2011*). Similarly, land-use change activities such as livestock farming, urbanization, and agriculture have been shown to cause declines in butterfly abundance and diversity (*Kruess & Tscharntke, 2002*; *Wallis De Vries et al., 2007*; *Casner et al., 2014*). However, because these studies do not integrate butterfly-plant interactions, they do not inform us about possible effects on the structure of their interaction networks. Indeed, compared to studies that consider bees or hummingbirds interacting with their plants, the effects of land-use change on plant-butterfly interaction networks remain largely unexplored (but see *Colom, Traveset & Stefanescu, 2021*).

Given the lack of sufficient knowledge about the effects of habitat conversion on plant-butterfly communities, this study had two main goals. The first aim was to evaluate how the conversion of a native juniper forest to anthropogenic land uses (especially agricultural and urban areas) in central Mexico affects the assemblages of butterflies and their flowers. The second aim was to evaluate changes in the structure of plant-butterfly networks from these areas. For our first aim, we expected a marked difference in the composition and diversity of butterfly and plant assemblages between human-modified areas and native juniper forest. This expectation is based on the previously observed trend of community composition change reported for both interacting biological groups in anthropized areas, wherein the richness of specialist species typically decreases, while that of generalist species increases (*Bergerot et al., 2011*; *Casner et al., 2014*). For the second aim, we thus expected interaction networks to be more generalist in anthropized areas, leading to a decrease in nestedness (*i.e.,* the pattern where the species with fewer connections interact with proper subsets of the species that are more connected) and an increase in modularity (*i.e.,* the extent to which species within the same module have more interactions with each other than with species in other modules) compared to the network from the native forest (*Hagen & Kraemer, 2010*; *Fortuna et al., 2010*; *Staniczenko, Kopp & Allesina, 2013*; *Payrató-Borràs, Hernández & Moreno, 2020*; *Colom, Traveset & Stefanescu, 2021*). Additionally, the connectance of the networks (*i.e.,* the proportion of realized interactions between entities in the network compared to the total number of possible interactions) in these areas may decrease as the number of potential interactions between plants and butterflies is reduced due to changes in habitat and species composition. Hence, the number of potential interactions in a butterfly-plant network could be diminished despite having a higher plant species richness. This could be attributed to factors such as behavioral preferences, timing of flowering and butterfly activity, or the absence of suitable nectar rewards. Furthermore, given their presence in the agriculture and urban areas, we expected alien plants to be strongly integrated into the interaction networks (*Maruyama et al., 2016*; *Parra-Tabla et al., 2019*), to the point that they may have a similar contribution to network structure as native plant

species. In Mexico, juniper forests occupy approximately 0.15% of the national territory, yet ecological knowledge of this forest type and its role in biodiversity conservation is precarious (*Aguirre-Calderón, 2015*; *Herrerías Mier & Nieto del Pascual, 2020*). Finding out the impact of the conversion of this native habitat on the interacting communities of butterflies and their plants will be a first step in highlighting the need for their study and conservation.

## MATERIALS & METHODS

### Study area

Butterfly-plant interactions were studied in an area measuring approximately 50 hectares in the municipality of Ixtacuixtla, located in the state of Tlaxcala in central Mexico (19°21′N and 98°22′W, with an altitude ranging from 2,300–2,350 m above sea level), from September 2021 to August 2022. The region has a mean annual precipitation of 900 mm, with a rainy season between June and September and a mean annual temperature of 13 °C (*Hudson et al., 2005*). The study area includes a gradient of different land-use categories. The original native vegetation is juniper forest (*Juniperus deppeana* Steud.) within a grassland dominated by *Muhlenbergia implicata* (Kunth) Trinius, *Stipa ichu* (Ruiz and Pavón) Kunth, *Aristida schiedeana* Trinius and Ruprecht (Poaceae), as well as *Rhus standleyi* F.A. Barkley (Anacardiaceae). Native forest has been historically replaced by urban settlements, where it is possible to find permanent ornamental gardens composed mainly of alien species. These settlements are surrounded by agricultural areas used mainly for corn and alfalfa crops, leaving some small preserved fragments of the original forest. Land use categories were determined through ground truthing, which involved physically visiting locations to validate land use through direct observations. These observations were then corroborated by satellite imagery of the sampled locations (*Olofsson et al., 2014*). In order to cover this gradient, we performed butterfly-plant interaction surveys at three sites (Fig. 1): Native forest (Native), agricultural lands (Agriculture) and the Universidad Autónoma de Tlaxcala Campus, to represent urban settlements (Urban). Because the land-use gradient is a continuum, we established a single 1-km-long transect at each site (3 kilometers in total). The distance between the end of one transect and the beginning of the next was approximately 100 m (Fig. 1). While the flying capacity of adult butterflies could allow for movement between the relatively close transects, the species' habitat preferences, behavioral tendencies (*e.g.*, territoriality and mating rituals), and the relatively smaller sampled area compared to their flying range suggest that the obtained results would still largely reflect the distinct butterfly communities within each specific habitat type.

### Surveys of plant-butterfly interactions

Censuses of plant-butterfly interactions were conducted twice a month, exclusively during clear and sunny conditions, from September 2021 to October 2022 on each transect. Two trained observers and one assistant steadily walked the transects performing the censuses during peak flight activity from 10:00 to 14:00 h. All butterfly surveys were conducted by the same observers. The decision to employ two observers for our butterfly surveys

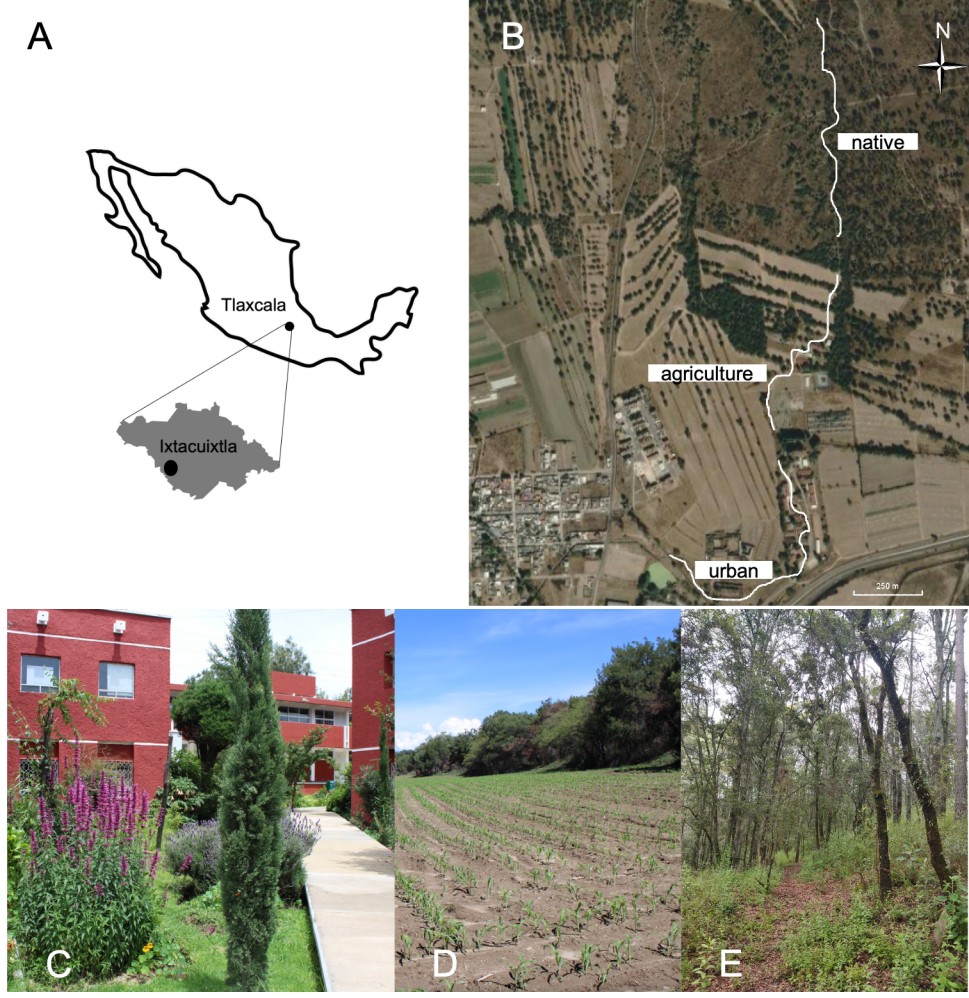

**Figure 1** **Map and satellite imagery from Google Earth™ showing the monitored locations.** (A) Geographic location of the study site in the state of Tlaxcala, Mexico. (B) Bold lines indicate the survey transects established to record plant-butterfly interactions in (C) urban, (D) agricultural and (E) native forest site.

was based on considerations related to survey accuracy, data quality, observer bias, and logistical feasibility. Through the study, the censuses started from a different transect and in a different direction to avoid order effects. Butterfly visits to flowers within 10 m on either side of the transect were counted, following the method of *Pollard & Yates (1993)*. A flower visit was defined as nectar-probing by a butterfly species, from the moment of proboscis insertion into the corolla to the end of proboscis withdrawal. When species identification by sight (using Eagle Optics binoculars) was difficult, butterflies were caught in a net, photographed (using a Canon EOS Rebel T7), and released. Samples of the plants with which the butterflies interacted were also collected. The specimens were identified using taxonomic keys for both the butterfly (*e.g.*, *Opler, 1994*; *Glassberg, 2007*) and plants (*e.g.*, *Calderón de Rzedowski & Rzedowski, 2005*).

## Richness and diversity of the interacting species

Butterfly and plant richness were expressed as the number of species present in each habitat type (*i.e.,* pooled across censuses over time). Both butterfly and plant richness included all interacting individuals recorded during all sampled surveys. To assess the impact of land use change on butterfly and plant diversity, we calculated the Shannon diversity index (H') with the identity and presence of butterfly and plant species per habitat type, and compared the values obtained using Hutcheson's robust *t*-test (*Hutcheson, 1970*). The percentage similarity test (SIMPER) was used to determine habitat dissimilarities in both butterfly and plant species composition and the main contributing species, using the S17 Bray–Curtis similarity matrix and a low contribution cut-off of 90%. This test was performed using the 'simper' function in the vegan package for R (version 2.4.4, *Oksanen et al., 2016*). The contribution of each species to the dissimilarity between samples is calculated based on both the species' abundance and their average dissimilarity. This dual consideration of abundance and dissimilarity ensures that species with significant differences in occurrence and abundance across samples are appropriately identified as key contributors to the dissimilarity. Sampling completeness of plant-butterfly interactions recorded in the study areas was determined using the 'iNext' function in the iNEXT package in R (*Chao & Jost, 2012*).

## Analysis of interaction networks

We summarized the interactions (*i.e.,* butterfly foraging visits to plants) of each plant-butterfly community in each habitat type using a bipartite matrix, where each cell indicated the frequency of pairwise interactions between a plant and a butterfly species. To compare these communities, we constructed three networks (original native juniper forest, agricultural land, and urban settlements) using all butterfly visits to plants as a measure of visitation frequency. We also calculated a quantitative measure of generalism for each network, which replaces the generality qualitative version obtained in the bipartite package (*Geslin et al., 2013*). This metric is defined as the number of plant species with which a given focal butterfly species interacted in the network (*i.e.,* links). The generalism measure took into account the link intensity, which represented the number of interactions for each link (*i.e.,* based on observations of butterflies on flowers). We used generalized linear mixed effect models (GLMM; *Bolker et al., 2009*) to analyse the effect of habitat type on the number of interactions at the network level and on the number of links per network. As numbers of interactions were count data, we fitted models with a Poisson distribution and a log link. The fixed effect was the habitat type, and the sampling week was included as a random effect. Similarly, the differences in the number of interactions recorded with native and alien plant species (response variable) in each type of habitat were analyzed using a GLMM. In this model, the type of habitat was treated as a fixed effect, and the sampling week as a random factor. Post-hoc test were performed using the glht function of the package multcomp (*Bretz, Hothorn & Westfall, 2010*). Pairwise comparisons between treatments were performed using Tukey's test.

Using the bipartite package (*Dormann, Gruber & Fründ, 2008*) in R software (*R Core Team, 2014*), we built and quantified the structure of these networks using a selected set

of network metrics, including connectance, vulnerability, modularity, and nestedness. Ecological network studies often examine these metrics, as they capture not only structural features but also ecological implications across organism and community scales. The first metric, connectance, is a measure of the density of connections in a network and was determined as the actual proportion of links of both butterflies and plants in the network compared to all possible links. The second, vulnerability, estimated the average number of links of butterflies per plant, a metric often used as an indication of the proportion of species likely to go extinct if one species in the network goes extinct (*Tylianakis, Tscharntke & Lewis, 2007*). The third, nestedness, is a metric of the degree to which species in a network interact in a non-random way. In a nested network, the interactions of the species with fewer interactions are a subset of the interactions of the species with more interactions. We used the normalized nestedness metric, NOD $F_c$, based on the NODF measure: $\text{NODF}_c = \text{NODF}_n/(C \cdot \log(S))$, where $\text{NODF}_n = \text{NODF}/\max(\text{NODF})$, $C$ is the connectance, $S$ is the geometric mean of the number of species in each level of the network, NODF is the raw NODF value for the network and max (NODF) is the maximum nestedness of a network with the same number of species and links as the focal network, subject to the constraint that every species has at least one link (*Song, Rohr & Saavedra, 2017*). This metric does not suffer from the statistical issues associated with $z$-scores and is thus robust for comparing nestedness between networks. It also controls for the impacts of connectance and network size on nestedness (*Song, Rohr & Saavedra, 2017*). We used the maxnodf package to implement calculation of the NOD $F_c$ metric (*Hoeppke & Simmons, 2021*). The fourth metric, Modularity (Q), is a measure of how the interactions in a network are structured, with high modularity indicating that interactions are concentrated within specific subgroups of nodes. This metric was estimated for both the qualitative and quantitative matrices. The QuaBiMo optimization algorithm (*Dormann & Strauss, 2014*), was employed for quantitative matrices. This algorithm was run independently 10 times, with the highest values selected. This iterative approach serves to enhance the algorithm's efficacy by facilitating the identification of optimal or near-optimal solutions, thereby enhancing solution quality and instilling greater confidence in the obtained results.

To assess the statistical significance of the network metrics (except nestedness), we compared the observed values to 1,000 random values that were generated using a randomization algorithm called Patefield's r2dtable algorithm, which conserves the total number of interactions per row and column in the matrix. This helps ensure that any observed differences between the observed metrics and the random values are not likely to occur due to chance alone. In other words, the null model is constructed in a way that reduces the probability of erroneously concluding that a significant difference exists when it doesn't. We expressed the network indices (connectance, and Q) as $z$-scores (observed - mean(null)/sd(null)), and evaluated the statistical significance using z-tests.

## RESULTS

### Land use-change affects the richness and diversity of interacting species

In the native forest site, we observed 25 butterfly species (Shannon diversity index, H' = 2.64, with nine butterfly species only found in this habitat) belonging to five families (Hesperiidae, Lycaenidae, Nymphalidae, Papilionidae, and Pieridae). In the agricultural site, we identified 20 butterfly species and 1 moth species (H' = 2.42, with two butterfly and 1 moth unique species to the site) belonging to six families (the five from the native forest site, plus Erebidae). In the urban site, we recorded 24 butterfly species (H' = 2.46, with 8 unique species to the site), which belonged to the same families found in the native forest site.

The Shannon diversity of butterfly species was not statistically different between the two types of anthropized vegetation (agriculture *vs.* urban, $t = 0.54$, *d.f.* = 774, $P = 0.58$). However, the Shannon index value of butterfly diversity in native forest was higher than that found in agricultural ($t = 3.21$, *d.f.* = 715, $P = 0.001$) and urban sites ($t = 3.08$, *d.f.* = 1040, $P = 0.002$), respectively.

In terms of plant species visited by butterflies during surveys, the native forest site had lower species richness than the agricultural and urban sites. The native forest site had 16 native plant species, nine of which were unique to this site. The agricultural site had 18 plant species, of which seven were unique to the site and two were alien species. The urban site had 44 plant species, of which 37 were unique to the site and 25 were alien species. Based on Hutcheson's robust *t*-test ($t = 12.88$ *d.f.* = 859, $P < 0.0001$), the Shannon diversity index differed significantly between agricultural land (H' = 2.15) and the urban site (H' = 3.05). Likewise, the plant species diversity was lower in the native forest site (H' = 1.36) than in the two modified sites (agriculture *vs.* native, $t = 10.89$, *d.f.* = 888, $P < 0.0001$; urban *vs.* native, $t = 23.97$, *d.f.* = 1050, $P < 0.0001$). The diversity and abundance of butterfly species and their associated plants recorded throughout the study in the three vegetation types are shown in File S1.

The average dissimilarities in butterfly assemblage composition in the sampled habitat types were 69.40% for the urban site *vs.* native forest, 63.09% for the agricultural *vs.* urban site, and 73.37% for the native forest *vs.* agriculture site. The main butterfly species that contributed to the observed differences between the urban site *vs.* native forest (34%) was *Leptophobia aripa*, which was mainly recorded in the urban site. When comparing the native forest *vs.* agricultural site (31%), it was *Danaus gilippus*, which was only recorded in the native forest. Between the agricultural *vs.* urban site (20%), it was *Ganyra josephina*, which was typically recorded in the agricultural site. In contrast, the average dissimilarities in plant assemblage composition were 98.48% for the urban site *vs.* native forest, 86.47% for the agricultural *vs.* urban site, and 81.73% for the native forest *vs.* agricultural site. *Bouvardia ternifolia*, a typical species from the native forest, was the main plant species that contributed to the observed differences between the native forest *vs.* agricultural site (99.2%). Meanwhile, *Lantana camara*, an ornamental species, contributed the most to the differences between the urban site *vs.* native forest (70.4%). Lastly, *Descurainia*

*virletii*, a common plant species found in crop fields, made the greatest contribution to the differences between the agricultural site *vs.* urban site (2.84%).

## Modification of native habitat impacts plant-butterfly networks

The observed number of butterfly-plant interactions in the study seemed to reach an asymptote in relation to our sampling effort across the three sampled sites (a total of 96 h of evenly distributed observation efforts among the sites/surveys throughout the study). We detected 99.67% of the interactions estimated for the native forest network, 99.74% for the agriculture network, and 99.81% of those estimated for the urban network according to the Chao2 estimator, after conducting 24 samples throughout the study.

The interaction matrices used to construct the networks of the three sampling sites are shown in File S2. Our results show variation in the network topology among sites. In the native forest site, we obtained 531 records of butterfly–plant interactions. The interaction network consisted of 41 species—25 butterfly species and 16 native plant species, with a total of 50 links (Fig. 2A). In the agricultural land site, we obtained 378 records of 21 butterfly species interacting with 18 plant species (2 of which were exotic species, Fig. 2B), and a total of 46 links. A total of 524 interactions were recorded in the urban site, involving 24 butterfly species and 44 plant species (25 of which were alien species, Fig. 2C), resulting in 83 links in total. Our GLMM showed differences between habitats in the mean number of links per network (Fig. 3A; $z = 2.94$, g.l. $= 2$, $P = 0.002$) as well as the mean number of interactions per network (Fig. 3B; $z = 14.28$, g.l. $= 2$, $P < 0.001$). More precisely, the mean number of links in the urban network was significantly higher than in the agriculture (Tukey test, $P = 0.044$) and native forest networks (Tukey test, $P = 0.002$), but the differences between these last two networks were not significant (Tukey test, $P = 0.589$). On the contrary, the native forest network had a higher mean number of interactions than the agriculture (Tukey test, $P = 0.001$) and urban networks (Tukey test, $P < 0.001$), which had similar level of interactions to each other (Tukey test, $P = 0.994$). Finally, the native plant species, particularly those present in the native forest, had a higher number of interactions in the study compared to the alien plant species that were present in both the agricultural and urban sites (Fig. 3C; $z = -2.22$, g.l. $= 61$, $P < 0.001$).

The native forest network had a significantly lower connectance (0.12, $z$-score $= -17.02$, $P < 0.001$), and significantly higher vulnerability (7.24, $t$-value $= 5.12$, $P < 0.001$). The $Q$ value (0.54, $z$-score $= -12.77$, $P < 0.001$) indicated significant modularity, which was higher than expected. That is, the network's organization into modules is more pronounced and structured than what one would typically observe due to random chance. This suggests that both butterflies and their plants being analyzed in the network are not randomly connected but are forming distinct and cohesive groups. A similar pattern was found for the agriculture network, which showed low connectance (0.12, $z$-score $= -21.95$, $P < 0.001$), low vulnerability (3.95, $t$-value $= -0.67$, $P = 0.503$), and statistically significant modularity ($Q = 0.56$, $z$-score $= -14.35$, $P < 0.001$). A significantly lower value of connectance (0.07, $z$-score $= -30.14$, $P < 0.001$) and vulnerability (2.28, $t$-value $= -8.18$, $P = 0.540$) was shown by the urban network.

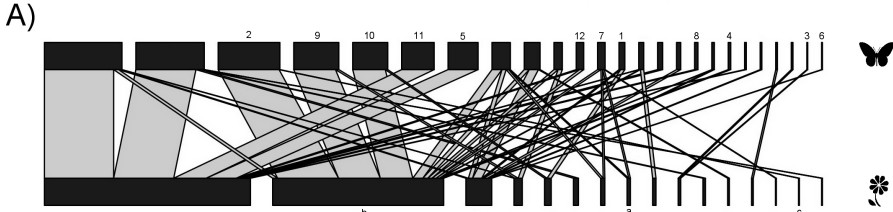

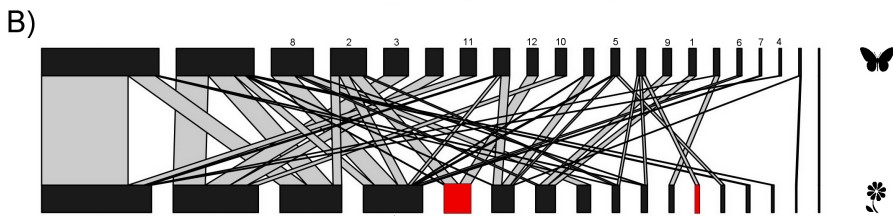

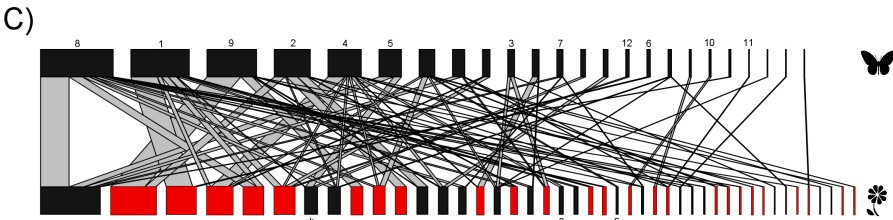

**Figure 2 Bipartite plant-butterfly interaction networks.** In each network the nodes on the top represent butterfly species, and on the lower side, different plant species. Red nodes represent alien plant species. The thickness of each link (gray lines) indicates the frequency of each pairwise interaction (butterfly flower visits). (A) Network for the native forest site. (B) Network for the agricultural site. (C) Network for the urban site. Same numbers and letters on the nodes denote shared species among the three study sites. Butterfly and plant species names that correspond to the codes for both numbers and letters are shown in File S2.

As with the networks from the other two vegetation sites, the modularity obtained for the urban network was significantly high ($Q = 0.61$, $z$-score $= -19.57$, $P < 0.001$). We obtained five and six modules for the native forest and agriculture networks, respectively. However, the urban network had 10 modules. The nestedness of the native forest network (NODF_c $= 2.63$) was higher than that of the agriculture network (1.91) and the urban network (2.29).

## DISCUSSION

Our findings revealed that the native forest site had a higher diversity of butterfly species than the agricultural and urban sites, with no significant difference observed between the two modified sites. However, the native forest had lower plant species richness and diversity compared to the two modified sites, particularly the urban site, which had a higher

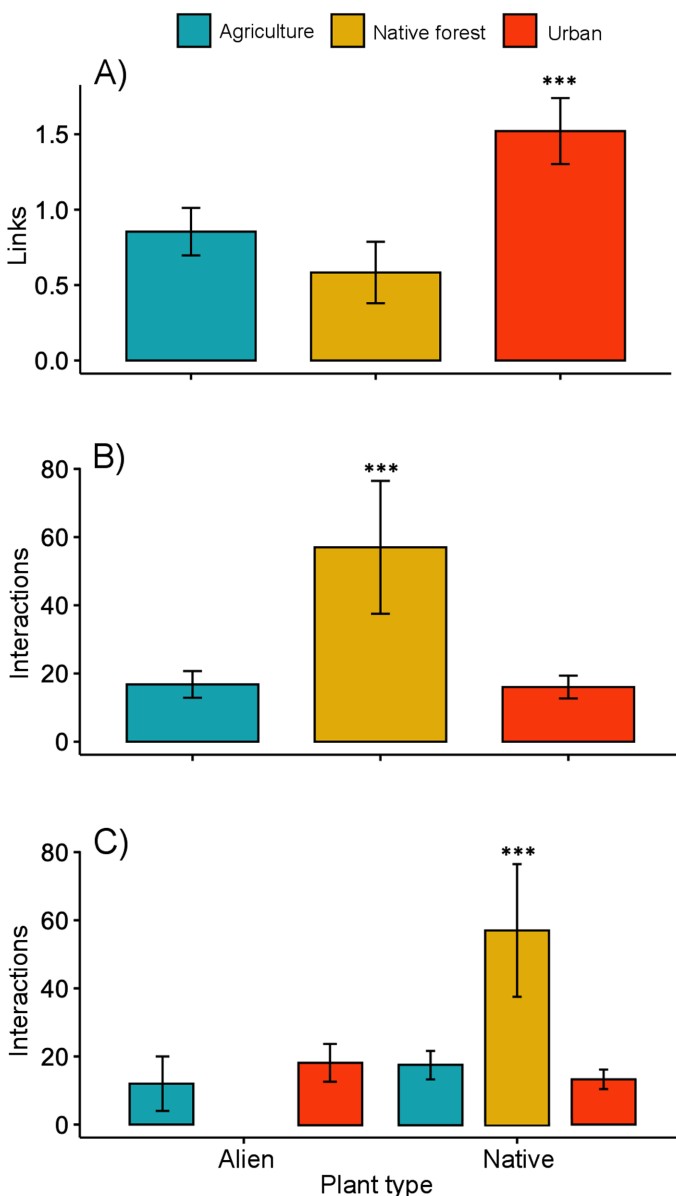

**Figure 3  Network level metrics by vegetation type (agricultural, urban and native forest).** (A) Mean number of links per network, (B) mean number of interactions between plants and butterflies, and (C) mean number of interactions between butterflies and native or alien plant species. Lines above bars denote ± standard errors. Asterisks above bars represent significant differences ($P < 0.001$).

richness of alien plant species. Among the sites, the species composition of both plants and butterflies exhibited variability. Interestingly, the dissimilarities in plant assemblages were mainly attributed to three native plant species, *Bouvardia ternifolia*, *Lantana camara*, and *Descurainia virletii*. Conversely, for butterfly assemblages, *Leptophobia aripa*, *Danaus gilippus*, and *Ganyra josephina* were the primary species contributing to dissimilarities

among the three habitats, respectively. These results suggest that anthropogenic activities such as agriculture and urbanization can have significant impacts on butterfly and plant diversity, leading to notable differences in network topology among the sites. The urban network had a significantly higher number of links than the agricultural and native forest networks, but the native forest network had a higher number of interactions. The native plant species—particularly those in the native forest—had a higher number of interactions than the exotic plant species. All three sites showed low connectance and significant modularity in their network structures, but the urban network had the highest number of modules compared to the native forest and agricultural networks. Moreover, the nested pattern within the native forest network exceeded that of both the agricultural and urban networks. Overall, our study revealed that land-use change not only had a significant impact on butterfly and plant diversity, but it also suggests that it could result in variations in the structure of interaction networks in disturbed sites when compared to the original forest. However, given the absence of replication or before/after surveys in our experimental design, the results must be approached with caution, as we mention later on.

Land use change has been identified as one of the major drivers of biodiversity loss worldwide, and flower visitors such as butterflies and their food host plants are no exception (*Bubová et al., 2015*). Butterflies are highly specialized in their choice of host plants, both for oviposition and for their feeding, making them vulnerable to habitat deterioration (*Erhardt & Thomas, 1991*; *Tiple, Khurad & Dennis, 2011*). Several studies have shown that land use change, particularly the conversion of natural habitats into agricultural or urban areas, can have negative impacts on butterfly diversity and richness (see *Öckinger & Smith, 2006*; *Wenzel et al., 2006*; *Kőrösi et al., 2012*). For example, *Maes & Van Dyck (2001)* reported significant changes in butterfly biodiversity in Flanders, with 69% of extant species declining due to urbanization and expansion of intensive agriculture. Similarly, studies in the Netherlands (*Van Dyck et al., 2009*), Sweden (*Franzén & Johannesson, 2007*), Finland (*Kuussaari et al., 2007*), and northeastern Spain (*Melero, Stefanescu & Pino, 2016*) have found declines in many butterfly species, particularly those that feed on grass or herbs in wetlands. In Europe, the declines have been most significant among specialist butterflies from grassland biotopes, wetlands and bogs, and woodlands/forests, due to habitat conversion into croplands and intensification of agricultural practices (*Van Swaay, Warren & Loïs, 2006*). A UK database found that 41 of 54 common butterfly species had declined since the 1970s, with 26% experiencing range decreases of over 40%; habitat fragmentation/destruction and intensification of agriculture were suggested as potential drivers of this loss (*Fox et al., 2006*). In the United States, butterfly populations have fluctuated due to habitat modification, particularly in prairie habitats and bogs (*Swengel & Swengel, 2016*). While some parts of Asia may have witnessed less severe declines in Lepidoptera populations, some of Japan's butterfly communities have been negatively affected by the gradual intensification of its traditional "satoyama" landscape (*i.e.,* rural landscape that combines agricultural areas with surrounding natural ecosystems; *Nakamura, 2011*).

All of the studies mentioned above are consistent with our findings, as butterfly diversity has been repeatedly shown to decline with increasing urbanization and agricultural intensification, while the diversity of plants (particularly exotic species) is favored in anthropized environments (*Walker et al., 2009*; *Raveloaritiana et al., 2021*; *Boscutti et al., 2022*). In our study, the highest plant diversity was observed in urban areas. The urban site we studied had many cultivated plants that served as food for butterflies and flowered throughout the year. However, some of these plants had shorter flowering periods, similar to those found in the native forest site. These nectar-producing plants, particularly alien species, benefited from artificial irrigation, fertilizer application, and other gardening activities. This provided a constant and varied source of nectar for the species of butterflies we recorded. For instance, *Autochton cellus*, a butterfly species that prefers moist and steep woodlands, was mainly found in the urban site, where it primarily feeds on the alien *Salvia officinalis*. This plant was widely available because residents plant it as an ornamental plant in their gardens. In addition, *Ganyra josephina*, a butterfly species that prefers open, dry, subtropical forests showed a preference for the agriculture site, where it mainly fed on *Descurainia virletii*, a typical weed found in crop fields. Interestingly, *Danaus gilippus*, a specialist species of open forests, was only found in the native forest site. This species primarily fed on the native plant *Verbena communis*. Our data highlight the potential for urban areas to support high plant species richness, even in comparison to natural areas. However, agricultural sites have often undergone some degree of biological homogenization due to the homogenization of resources within the physical environment (*Solar et al., 2015*). These effects also impact the structure of the interaction networks between plants and butterflies, as we discuss below

In a plant-flower visitor interaction network, a greater link intensity (*i.e.,* a higher number of interactions) indicates more intense and specialized interactions between particular species, which can have implications for the stability and structure of the network. In our study, networks from the three sites had low connectance, suggesting that butterflies strongly interact with few plant species in each habitat type (*Ibanez, 2012*). The number of links was higher in the urban site, suggesting that an increase in non-native plant richness in this site promotes an increase in links with generalist butterfly species. A large number of interactions for each link in a plant-flower visitor network suggests a network that is more complex, diverse, and resilient, as suggested by the vulnerability indicators in our study. This complexity emerges from the interactions among the species involved (*Lopezaraiza-Mikel et al., 2007*). All of our networks showed nested and modular structures, although nestedness was highest for the network of the native forest site. However, in the urban site, the network showed a higher number of modules than the others. The presence of both structural patterns in all three networks suggests that there is a combination of specialized nodes that interact with generalist nodes, as well as highly connected nodes that cluster into communities. This structure may be important for understanding how interactions between species are organized and how disturbances or changes in the environment could affect the stability of the network. Thus, our results suggest that land-use change directly impacted the structure of the plant-butterfly networks, which have only been minimally documented in the

literature. For example, some studies have explored the effects of different agricultural practices on plant-butterfly networks and identified a trend where specialized interactions are the first to disappear (*Aizen, Sabatino & Tylianakis, 2012*). Such losses can both decrease nestedness and increase modularity, decreasing network stability (*Spiesman & Inouye, 2013*; *Colom, Traveset & Stefanescu, 2021*). In our study, we observed these patterns, where a land-use change from native forest resulted in a decrease in nestedness in agricultural and urban site networks, but an increase in their modularity. There are several possible explanations for this trend. For example, it is possible that this is due to the loss in agricultural and urban areas of the substantial number of plant and butterfly species that previously interacted in native forests. Or, the notable increase in plant species in the urban site could promotes greater opportunities for generalist feeding habits in the butterflies. Moreover, the few studies conducted so far have shown that the level of specialization in plant-butterfly networks decreases in urban areas compared to forested or agricultural areas (*Baldock et al., 2015*), with a tendency towards a greater diversity of generalist butterfly species (*Geslin et al., 2013*). This pattern has been attributed to the richness of alien plants in urban sites, which attracts generalist butterflies (*Lopezaraiza-Mikel et al., 2007*). However, despite the increase in the number of plants with which a given butterfly species interacts in the network (number of links) in the urban site, there is a decrease in butterfly species richness associated with impervious surfaces (*Herrmann, Buchholz & Theodorou, 2023*). Our study is consistent with these findings, since despite the high richness of alien plants in the urban site, a smaller number of butterfly species were observed compared to the native forest.

In order to properly understand and assess the impacts of land use change and the introduction of exotic plants on the structure of plant-butterfly networks, more research is needed to identify the mechanisms by which these changes occur. For instance, future studies should focus on determining how different types of land use impact different butterflies species and their interactions with native plants. In addition, research should be directed at identifying the most effective methods for conserving and restoring native populations and interactions when exotic species are present.

Finally, it is important to note that changes in land use and the introduction of exotics do not always have negative impacts on plant-butterfly networks. For example, some alien plants may provide resources such as nectar or pollen that native species are unable to provide, thus increasing the diversity of these networks (*Kremen et al., 2007*). Therefore, it is essential to consider both the positive and negative impacts of these changes when planning conservation and management strategies to protect and restore native forested areas.

## STUDY LIMITATIONS

Our study analyzed butterfly and plant communities in three distinct habitat types: native forest, agricultural land and urban settlements. Differences in community composition and plant-butterfly interactions were found among these habitats. However, a significant limitation exists due to the lack of true replication in the study design. Each land use

category has only one type of vegetation, posing a challenge to result robustness. Without multiple instances of each habitat, it is difficult to distinguish whether differences arise from land use variations or unique vegetation attributes. Moreover, using random effects in our statistical models may lead to pseudoreplication, as single instances of each habitat are treated as independent replicates. Likewise, our study uncovers intriguing variations in network topology among sites. Despite the fact that the networks do not vary significantly in terms of network size or the number of species, the interpretation of these results should be approached with a nuanced understanding of how network size can influence measures such as connectance, vulnerability, and modularity. Considering the complex interplay between network size and these measures is crucial for drawing accurate ecological conclusions and making valid comparisons across networks of differing sizes. In conclusion, though the study offers valuable insights into diverse butterfly and plant communities across land uses, limitations in replication and potential pseudoreplication, and a network size effect highlight the need for cautious interpretation. Future research with proper replication, experimental design, and accounting for the potential influence of network size is crucial for validating and refining these findings.

## CONCLUSIONS

Our study provides valuable insights into the effects of land use change on butterfly communities and their associated plants, as well as the resulting alterations in interaction networks. Thus, our findings emphasize the importance of native forest conservation in mitigating biodiversity loss, particularly for specialized butterfly species and native plant communities. Land use change, especially urbanization and agricultural intensification, poses significant threats to butterfly diversity and the integrity of ecological networks. To promote effective conservation and management strategies, further research is required to understand the mechanisms driving these changes and the potential impacts of exotic plant species on native interactions. In the face of ongoing environmental challenges, a comprehensive approach that considers both positive and negative impacts of land use change will be crucial for preserving and restoring natural ecosystems.

## ACKNOWLEDGEMENTS

We gratefully acknowledge Mauro Piedras for field assistance. Two anonymous reviewers provided useful comments on previous versions of the manuscript. Thanks to Lynna Kiere for language editing. Three anonymous reviewers made useful comments on previous versions of the manuscript. This work constitutes partial fulfillment of D. M. G.'s master degree requirements at UATx.

### Funding

This work was supported by the Consejo Nacional de Humanidades, Ciencia y Tecnología (CONAHCyT) through a masters' scholarship (Number: 1143880) to Deysi Muñoz. The

funders had no role in study design, data collection and analysis, decision to publish, or preparation of the manuscript.

## Grant Disclosures

The following grant information was disclosed by the authors:
The Consejo Nacional de Humanidades, Ciencia y Tecnología (CONAHCyT): 1143880.

## Competing Interests

The authors declare there are no competing interests.

## Author Contributions

- Deysi Muñoz-Galicia conceived and designed the experiments, performed the experiments, analyzed the data, prepared figures and/or tables, authored or reviewed drafts of the article, and approved the final draft.
- Carlos Lara conceived and designed the experiments, performed the experiments, analyzed the data, prepared figures and/or tables, authored or reviewed drafts of the article, and approved the final draft.
- Citlalli Castillo-Guevara analyzed the data, prepared figures and/or tables, authored or reviewed drafts of the article, and approved the final draft.
- Mariana Cuautle analyzed the data, prepared figures and/or tables, authored or reviewed drafts of the article, and approved the final draft.
- Claudia Rodríguez-Flores analyzed the data, prepared figures and/or tables, authored or reviewed drafts the article, and approved the final draft.

## Data Availability

The raw measurements are available in the Supplementary Files.

## Supplemental Information

Supplemental information for this article can be found online at http://dx.doi.org/10.7717/peerj.16205#supplemental-information.

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
