# Peer review of "Impacts of land use change on native plant-butterfly interaction networks from central Mexico"

_PeerJ, doi:10.7717/peerj.16205_

## Round 0.1 · original submission · Major Revisions

Dear Dr. Lara,

After this first review round, two reviewers indicated the need for a major review, while a third reviewer indicated the need for a minor review. I consider that all criticisms raised by the reviewers need to be considered before your text is accepted for publication in PeerJ. Therefore, I invite you to perform the changes required by the reviewers and prepare a rebuttal letter informing the changes that were implemented and those that were not accepted in the new version of your text.

After your manuscript receives all the improvements indicated by the reviewers, I am sure your manuscript will be closer to a final acceptance.

Sincerely,
Daniel Silva

Reviewer 1 ·

Basic reporting

See my review in Section 4.

Experimental design

See my review in Section 4.

Validity of the findings

See my review in Section 4.

Additional comments

Review of "Impacts of land use change on native plant-butterfly interaction networks from central Mexico" by Muñoz-Galicia et al.

The authors study the network structure of interactions among plants and their butterfly pollinators in three land-uses in central Mexico: juniper forest, urban settlements, and agricultural land. They sample data twice per month for one year and build three networks representative of each land-use. They describe changes in species diversity and various network structure properties (including connectance, nestedness, and modularity) across the environmental gradient, finding that the urban network has the highest number of network links and modules.

The manuscript is well-written, the methods are generally sound, and the results are interesting and nicely discussed. I am generally supportive of publication if the below changes are implemented. I describe my suggestions in the order they appear in the text.

-- L52. The authors need to justify their final statement in the Abstract since it doesn't follow obviously from the preceding sentences.

-- L68. The authors should include reference to work on host-parasitoid networks that have the same formal structure as plant-pollinator networks. Two relevant references that involve host-parasitoid networks and land-use change are:

Tylianakis et al. (2007). Habitat modification alters the structure of tropical host–parasitoid food webs. Nature, 445, 202–205.

Staniczenko, P.P.A. et al. (2017). Predicting the effect of habitat modification on networks of interacting species. Nature Communications, 8, 792.

-- L78. Another useful, recent reference is:

Baldock, K.C.R. et al. (2019). A systems approach reveals urban pollinator hotspots and conservation opportunities. Nature Ecology & Evolution, 3, 363–373.

-- L106. The authors should include introductory descriptions of modularity and nestedness and some relevant references, such as:

Fortuna, M.A. et al. (2010). Nestedness versus modularity in ecological networks: two sides of the same coin? Journal of Animal Ecology, 79, 811–817.

Staniczenko, P.P.A. et al. (2013). The ghost of nestedness in ecological networks. Nature Communications, 4, 1931.

Payrató-Borràs et al. (2020). Measuring nestedness: A comparative study of the performance of different metrics. Ecology and Evolution, 10, 11906–11921.

-- L108. Define connectance.

-- L197. There are well-known issues with early implementations of the NODF and wNODF methods. I encourage the authors to redo analyses with the current best-practice methods for measuring nestedness, as described in Payrató-Borràs et al. (2020).

-- L253 to L260 specifically, and also throughout the text. The authors should clarify the distinction between "interactions" and "links" and make sure the use of the two terms is consistent throughout the manuscript to remove any confusion, e.g. L207. It would also be helpful to add values for the number of links in each network in Results.

-- Results. I recommend the authors analyse two additional network structure metrics, generality and vulnerability, which sit between connectance and nestedness/modularity in terms of complexity. Two relevant references (the first for a general introduction to generality and vulnerability and the second for an implementation for bipartite networks) are:

Bersier et al. (2002). Quantitative descriptors of food-web matrices. Ecology, 83, 2394–2407.

Tylianakis et al. (2007). Habitat modification alters the structure of tropical host–parasitoid food webs. Nature, 445, 202–205.

-- Results. I suggest adding a table that, for each network, summarises: #plants, #pollinators, #interactions, #links, connectance, generality, vulnerability, modularity, nestedness (and any significance values). This would provide a straightforward way of comparing the differences and trends between networks and land-uses.

-- L285. Move this statement to the beginning of Results.

-- Discussion. The Discussion is excellent but does include some very long paragraphs---consider breaking them up into shorter blocks of text focused on a single key idea.

Reviewer 2 ·

Basic reporting

The manuscript “Impacts of land use change on native plant-butterfly interaction networks from central Mexico” by Muñoz-Galicia et al compares interactions between flower-visiting Lepidoptera and flowering plants at three sites over the course of a year in central Mexico. Thank you to the authors for the work they put into collecting and analyzing the data from an understudied system. In general, there is a lot to commend this manuscript.
However, my greatest concern is the lack of replication. With each habitat type (land use) represented by a single site, and all within close proximity of one another, the ability to explain the cause of any differences between the sites is extremely impaired. The authors use the repeated measures sampling as their unit of replication throughout, but this is not a true unit of replication for the core study question, which is framed around land use and its effects on the interactions between butterflies (and one moth family) and plants. Alternative ways to assess the impact of land use change, as it is framed in the manuscript, might be such surveys before and after land use change. In this case, we lack a clear understanding of the time since this land has been altered for anthropogenic purposes, and lack spatial replication that would indicate true variance among land use types.
The simplest way to address this would be to be less ambitious with the framing and interpretation of the study. Rather than suggesting that the differences between these sites represent the impact of land use change more broadly, it could be framed as a case study in a unique ecosystem, with a year-long survey in temporal variation in these networks. In my opinion, the regular sampling twice each month over the duration of the entire year is the most interesting and valuable aspect of this study and I would rather the authors focus on that temporal dynamic rather than trying to focus on the land use, for which they have much weaker evidence.
Regardless of the direction the authors choose to move forward with this study, it is their responsibility to make clear the limitations of the sampling, and to state explicitly that the random effects used in the models represent pseudoreplication/repeated measures.

Experimental design

1) How is the land use gradient quantified? Did the authors measure land use at a spatial radius or what other method is used to define the different land use categories?
2) This is nitpicky, but it probably isn’t appropriate to assume the butterflies/moths are acting as pollinators. Might be better to call them “flower visitors” throughout
3) Links and interactions are often used interchangeably in network literature. I recommend using “frequency” or “abundance” in place of “interactions” here.
4) Several of these network measures (e.g. connectance) are strongly influenced by network size/the number of species in the network. How does this affect your interpretation of the results?
5) The relationship between generality/connectance/nestedness/modularity are not clear throughout. Why would increasing generality increase modularity? One would assume that generalists would connect the modules. How does generality affect nestedness? Why not directly measure the specialization of the species or the network as a whole?

Validity of the findings

I do not believe that the current framing of the manuscript appropriately interprets the data collected. The models are pseudoreplicated based on repeated measures, and there is no quantitative land use gradient established.

Additional comments

Line by line notes:
Abstract: Line 32: Explicitly state the number of sites surveyed per land use type.
Line 37—39: The description of unique species is unclear.
Line 46-47: How many modules?
Line 51-52: What is the biological interpretation of this result?
Line 53: Are the results described here a negative effect? Why?
Intro: Line 73-75: It is not clear how this line follows from the previous. Why would the disappearance of specialists lead to an increase in modularity?
Line 76: I think this is actually a relatively well-studied field, the long list of citations on lines 79-80 doesn’t support the statement that it is poorly studied.
Line 81: Many organisms depend on plants throughout their lives, I’m not sure butterflies are exceptionally dependent compared to others.
Line 84: The mutual dependence of the plants on butterflies as pollinators is not established here.
Line 104: Was this directly tested? Could it be?
Line 106: Not sure whether this logic holds
Line 109- 110: Why would the number of potential interactions be reduced if the plant species richness increases? What is the relationship between connectance and network size?
Methods: Line 127: How was this gradient quantified?
Line 138-139: How does the close proximity of these sites potentially affect the results? The adult butterflies can easily move between sites, right? And the larval butterflies were not measured here
Line 142: How is a mild weather day defined?
Line 143: How is the number of observers determined? Is this consistent among sites/surveys?
Line 153-154: Keys and taxonomists should be cited/named
Line 157: Pooled across censuses over time?
Line 173-174: Perhaps visitation frequency instead of interaction strength, which has a precise meaning
Line 174: this is a quantitative not qualitative measure
Line 175: It’s not clear whether there is one focal butterfly species or whether this is averaged across species
Line 177: Is “link intensity” just abundance? Are there background measures of plant or butterfly abundance or is this based on observations of butterflies on flowers?
Line 203-204: I’m not familiar with this method, why choose the highest values from 10 independent runs?
Line 208: What does it mean that the “Null model is not susceptible to type I errors.”?
Results: Line 218: Erebidae is considered a moth family
Line 221: Shannon diversity or species richness?
Line 225: Better to be consistent in the terminology here… sometimes this is called the native vegetation site and other times the native forest site
Line 229-231: This abundance is based just on the plants that were visited by butterflies during the survey correct? Or was there a separate plant diversity survey?
Line 238-240: I don’t understand this section about how a single species was contributing most to the distance matrices. Is this based on abundance? If so, I would be explicit about it.
Line 272: Why include two separate measures of NODF?
Line 274: Interpret the modularity value for the reader.
Line 284: Does 10 modules make sense based on the network size here? Can you draw the modules on the figure?
Line 286-287: This should be in the methods. Also, is this evenly distributed among the sites/surveys?
Discussion: Lines 292-293: I don’t think this opening sentence is supported by this study.
Lines 296-298: One thing you could measure is the asymmetry in butterfly/plant species in each network
Lines 206-207: I don’t think the authors can establish this based on this study, without replication or before/after surveys. Certainly cannot be established that the land use is causal for the differences in these networks
Line 319: Only adult behavior, not larval feeding, was measured here
Lines 322-339: This should be in the introduction, not discussion
Line 342-343: Cite specific papers for these specific results
Lines 343-352: This should be in the introduction, not discussion
Lines 369-370: This was not evaluated here
Line 373: This sentence is unclear
Line 375: Why not directly measure specialization?
Line 408: Functional homogenization was not measured here
Line 409: Was abundance measured here?
Conclusions: I suggest keeping the conclusion paragraph closer to the work presented. This is too broad for this study.
Lines 439-440: This wasn’t shown in this paper.
Figures: Figure 2 caption “and on the lower side” instead of “on the down”
Figure 2: Can you show the modules here?
Figure 3 caption: Spell out “standard error”
Figure 3: The land use is called a gradient throughout but it is not clear what the gradient is here. From agricultural to urban? Or from anthropogenic to natural? There is not a gradient on the x-axis that I can see.
Would be nice to have a figure showing the dissimilarity/unique species.

Reviewer 3 ·

Basic reporting

This paper presents an interesting investigation into the effects of land-use change upon plant-lepidopteran interaction networks. The study is clear and well written, the introduction gives an accurate theoretical background to the investigation and propose an interesting hypothesis to test.

Experimental design

The study was performed in a rigorous way, they use standard methods of butterfly sampling and sampled their sites several times during a whole year in order to get a good representation of species interactions. The methods are clearly described so it can be replicated.

Validity of the findings

The results are clear, the graphics are well presented and they provide enough information to interpret them. However, the study was only conducted in three sites, each one representing one habitat type. Therefore, it is hard to attribute the differences between sites to land-use type only since there is no replication. I understand the difficulty of finding these replicates, however this shortcoming should be addressed in the discussion. Why do you think these three sites are representatives of the land-use types? You need to strengthen your argument.

In the same line of thinking, in lines 221-224 you describe the statistics to compare the diversity between sites, however the degrees of freedom you report seem too many, as if each sample time was independent, but you have to consider that the sampling times are all taken in only 3 sites, therefore you do not have that many degrees of freedom. The way you are reporting the statistical analysis seems to have temporal-pseudoreplication, please revise.
I recommend to review this paper
Heffner, R.A., Butler, M.J., & Reilly, C.K. (1996). Pseudoreplication revisited. Ecology, 77(8), 2558-2562. doi: 10.2307/2265754

Additional comments

L123. Please insert “N” after 19º21´
L307. “number”
L383. Why don´t you calculate the robustness indicators so you can strengthen this argument?

Figure 2. It would be interesting to highlight the shared species between networks; this could be done with letters, numbers, or colors.

---

## Round 0.2 · Minor Revisions

Dear Dr. Lara,

Congratulations! Please consider your manuscript accepted in practice. Still, I provided your manuscript a minor review status for you to proceed with very minor suggestions and corrections made by reviewer #2. Please proceed with the changes and as soon as you resubmit the manuscript will be formally accepted for publication in PeerJ.

Sincerely,
Daniel Silva

Reviewer 1 ·

Basic reporting

See 4.

Experimental design

See 4.

Validity of the findings

See 4.

Additional comments

Review of revised "Impacts of land use change on native plant-butterfly interaction networks from central Mexico" by Muñoz-Galicia et al.

I am Reviewer 1 from the first round of reviews. I have read the authors' response and the revised manuscript---I am satisfied with the updates and I am supportive of publication.

Reviewer 2 ·

Basic reporting

I was one of the reviewers on an earlier draft and am overall happy with the revised draft. Thank you to the authors for their hard work. The new figure showing the sites is very helpful.

I have a couple minor notes (below).

Experimental design

The authors have addressed my concerns regarding the interpretation of the results.

Validity of the findings

The authors address the limitations of their study design in the discussion.

Additional comments

(using the tracked changes "all markup" version for line numbers)
Minor comments:
Abstract line 36: per land use type? and not sure what "interaction were summarized the matrices for each habitat" ... maybe interactions were summarized in matrices on which we calculated network descriptors?
Line 69: Butterfly is misspelled
Introduction line 111: more closely related may imply phylogenetic relatedness, perhaps more tightly connected? or more likely to interact?

Reviewer 3 ·

Basic reporting

No comment

Experimental design

No comment

Validity of the findings

No comment

Additional comments

No comment

---

## Round 0.3 · accepted · Accept

Dear Dr. Lara,

I am pleased to inform you that your manuscript has been formally accepted for publication in PeerJ!

Sincerely,
Daniel Silva